# Driver Genetic Mutations in Spinal Cord Gliomas Direct the Degree of Functional Impairment in Tumor-Associated Spinal Cord Injury

**DOI:** 10.3390/cells10102525

**Published:** 2021-09-24

**Authors:** Yoshitaka Nagashima, Yusuke Nishimura, Fumiharu Ohka, Kaoru Eguchi, Kosuke Aoki, Hiroshi Ito, Tomoya Nishii, Takahiro Oyama, Masahito Hara, Yotaro Kitano, Hirano Masaki, Toshihiko Wakabayashi, Atsushi Natsume

**Affiliations:** 1Department of Neurosurgery, Nagoya University, Nagoya 466-8550, Japan; nagashima4251@gmail.com (Y.N.); fohka@med.nagoya-u.ac.jp (F.O.); kaoki@med.nagoya-u.ac.jp (K.A.); diamonddust.h@gmail.com (H.I.); tomoya.nishii@gmail.com (T.N.); oyama.takahiro.0504@gmail.com (T.O.); youtarou0203@yahoo.co.jp (Y.K.); mahirano@med.nagoya-u.ac.jp (H.M.); wakabat@med.nagoya-u.ac.jp (T.W.); 2Department of Neurosurgery, National Hospital Organization Nagoya Medical Center, Nagoya 460-0001, Japan; eguchi.kaoru.nougeka@gmail.com; 3Institutes of Innovation for Future Society, Nagoya University, Furo-cho, Chikusa-ku, Nagoya 464-8603, Japan; 4Department of Neurosurgery, Aichi Medical University, Nagakute 480-1195, Japan; masahara219@hotmail.co.jp; 5Department of Neurosurgery, Mie University, Tsu 514-8507, Japan; 6Division of Cancer Biology, Aichi Cancer Center, Nagoya 464-8681, Japan; 7Division of Neurosurgery, Kawamura Medical Society Hospital, Gifu 501-3144, Japan

**Keywords:** glioma, spinal cord astrocytoma, diffuse midline glioma, genetic, mutation, H3F3A, H3K27M, IDH

## Abstract

Genetic analysis in glioma has been developed recently. Spinal cord glioma is less common than intracranial glioma. Thus, the clinical significance of genetic mutations in spinal cord gliomas remains unclear. Furthermore, because the spinal cord is an important communication channel between the brain and the rest of the body, increased attention should be paid to its functional prognosis. In this study, we investigated the functional prognosis and driver genetic mutations in eight patients with spinal cord gliomas (World Health Organization grade I, three cases; grade II, two cases; grade III/IV, three cases). *IDH* mutations were detected in all grade II cases and *H3F3A* mutations were detected in all grade III/IV cases. The functional status of grade I and II gliomas remained unchanged or improved 1 year after surgery, whereas grade III/IV gliomas remained unchanged or deteriorated. Spinal glioma progenitor cells with *H3F3A* mutations were associated with accelerated tumor-associated spinal cord injury, which led to functional impairment. Conversely, the presence of *IDH* mutations, which are rarely reported in spinal gliomas, indicated a relatively favorable functional prognosis.

## 1. Introduction

Intramedullary spinal cord gliomas are rarely encountered in central nervous system (CNS) tumors [1,2]; however, they frequently lead to severe neurological deterioration. According to a study that analyzed 1033 newly diagnosed spinal glioma cases between 2004 and 2016 based on the National Cancer Database in the United States, World Health Organization (WHO), grade I pilocytic astrocytomas, accounting for 19% of intramedullary spinal gliomas, largely enable maximal safe surgical resection, leading to better outcomes [3]. In contrast, WHO grade II and III astrocytomas (52%) that longitudinally infiltrate the spinal cords are less amenable to safe surgical resection. WHO grade IV astrocytoma (glioblastoma, 29%) requires adjuvant radiation and chemotherapy as it is the most devastating to the spinal cord. Given the premium on preserving neurologic function during spinal cord surgery, tumor types play an important role in the degree of spinal cord injury caused by the nature of the tumor and surgical difficulty. Recent genome-level sequencing studies of infratentorial gliomas revealed that discrete tumor types have tumor progenitor cells (i.e., cancer stem cells) with specific genomic driver alterations. Several genetic alterations are thought to play pivotal roles in tumorigenesis.

Histone H3 K27M mutations in genes encoding H3.3 (*H3F3A*) or H3.1 (*HIST1H3B/C*) are frequently found in high-grade infiltrative gliomas arising in the brain stem, thalamus, and spinal cord [4,5]. A small group of spinal cord astrocytomas harbors isocitrate dehydrogenase (*IDH*) mutations, namely, *IDH1* or *IDH2* mutations [6,7]. In contrast, *BRAF-KIAA1548* translocations are the most recurrent findings in WHO grade I spinal astrocytomas, but they are not the founder genetic alteration [8].

Although surgeons make every effort to preserve neurologic functions, no study has investigated how the divergence of mutated progenitor cells could deteriorate or recuperate neurologic functions. This study with a cohort of eight spinal gliomas sought to examine which types of tumor progenitor cells accelerate functional impairment in tumor-associated spinal cord injury.

## 2. Materials and Methods

### 2.1. Patients

Eight patients with spinal glioma underwent surgical removal between 2016 and 2019 (Table 1). The tumor tissues were used for analysis in this study, with informed consent from the Nagoya University. One patient was 14 years old at the time of diagnosis, three were adolescent young adults (age ≥ 15 years and ≤39 year), and the remaining four were over 40 years old. This cohort consisted of three pilocytic astrocytomas, two astrocytomas with *IDH* mutations, and three grade III/IV astrocytomas with the *H3F3A* mutations. The location and extent of resection for each tumor and the change of neurological status before surgery, 1week- and 1 year after surgery are described in Table 1 and Table 2, respectively. The concrete medical histories in Cases #4 and #6 are presented in the Results section.

### 2.2. Genetic Analysis

DNA was extracted from frozen tumor samples from all cases, using the QIAmp DNA Mini Kit (Qiagen, Hilden, Germany) according to the manufacturer’s instructions. We performed Sanger sequencing to detect *IDH1* and *H3F3A* mutations in DNA extracted from tumor samples. We amplified 129 and 194bp fragments spanning the sequence encoding the catalytic domain of IDH1, including codon 132 and the histone H3 lysine (K) 27 of the *H3F3A* gene, respectively. The conventional PCR conditions and primers were as follows: for *IDH1*, 35 cycles of denaturation at 95 °C for 30 s, annealing at 56 °C for 40 s, and extension at 72 °C for 50 s, with a final extension step at 72 °C for 7 min, using the forward primer 5′-CGGTCTTCAGAGAAGCCATT-3′ and reverse primer 5′-GCAAAATCACATTATTGCCAAC-3′; for *H3F3A*, 35 cycles with denaturation at 98 °C for 10 s, annealing at 55 °C for 30 s, and extension at 68 °C for 30 s, with a final extension step at 68 °C for 5 min, using the forward primer 5′-TGCTGGTAGGTAAGTAAGGAG-3′ and reverse primer 5′-AGCAGTAGTTAAGTGTTCAAATG-3′. Direct sequencing was performed using the BigDye Terminator v1.1 Cycle Sequencing Kit (Applied Biosystems, Foster City, CA, USA). The reactions were performed using an ABI 3100 Genetic Analyzer (Applied Biosystems) [9]. As immunohistochemistry for the anti-IDH1 R132H antibody (H09, Dianova, Hamburg, Germany) showed a faint positivity in Case #5, we performed droplet digital PCR (ddPCR) as described previously [10]. The reaction mixture (20 μL) for ddPCR comprised 10 μL of 2× ddPCR Supermix for Probes (BioRad Laboratories, Pleasanton, CA, USA), 1.8 μL of forward primer (10 pmol/μL), 1.8 μL of reverse primer (10 pmol/μL), 0.5 μL of VIC-labeled WT probe (250 nM), and 0.5 μL of FAM-labeled R132H probe (250 nM) with 1, 10, and 30 ng of DNA from each tumor specimen. The forward primer was 5′-CTTGTGAGTGGATGGGTAAAACCTA-3′ and the reverse primer was 5′-CCAACATGACTTACTTGATCCCCATA-3′. Fluorescent TaqMan MGB probes (FAM-labeled *R1322H* mutant-5′-ATCATAGGTCATCATGC-3′ and VIC-labeled WT-5′-CATCATAGGTCGTCATGC-3′) were synthesized by Thermo Fisher Scientific (Waltham, MA, USA) [11]. Each mixture was mixed with 70 μL of Droplet Generation Oil (Bio-Rad, Hercules, CA, USA), and droplets were generated using a QX200 Droplet Generator (BioRad). PCR amplification was performed using a C1000 thermal cycler (BioRad).

### 2.3. Neurological Assessment

To evaluate clinical outcomes in patients who underwent surgical resection for spinal gliomas, neurological symptoms including impairment of motor, sensory, and bladder function were evaluated before and at 1 week and 1 year after surgery based on the medical records. To quantitatively compare the neurological status of the patients, the American Spinal Injury Association (AISA) impairment scale (AIS) and the Modified McCormick Scale (MMS) were also measured. AIS is the most widely used neurologic classification of spinal cord injury and the scale divides spinal cord injuries into 5 categories. (A = complete injury; no sensory or motor function is preserved in the sacral segments S4–S5, B = incomplete injury; sensory but not motor function is preserved below the neurological level and includes the Sacral Segments S4–S5, C = incomplete injury; motor function is preserved below the neurological level and more than half of key muscle functions below the neurological level of injury have a muscle grade less than 3 (Grades 0–2), D = incomplete injury; motor function is preserved below the neurological level and at least half of key muscle below the neurological level have a muscle grade ≥ 3, E = normal; sensory and motor function are normal.) [12]. The MMS is a grading system that focuses on gait function and is often used to measure postoperative outcome in intramedullary spinal cord tumors [13,14,15]. The MMS is based on the patient’s motor function and sensory impairment status (I = normal ambulation; II = mild motor or sensory deficit, independent without external aid; III = moderate deficit, limitation of function, independent of external aid; IV = severe motor or sensory deficit, limited function, dependent; and V = paraplegia or quadriplegia, even with flickering movement).

### 2.4. Radiological Assessment

Radiological assessments were independently performed by two spine surgeons certified by the Neurospinal Society of Japan. Tumor location and size, before and after surgery, were compared using magnetic resonance imaging (MRI). Postoperative radiographic evaluation was performed within 1 month after surgery.

## 3. Results

### 3.1. Comparison of Spinal Glioma Cases with the IDH1 and H3 K27M Mutations

A 53-year-old woman (Case #4) with a 2-year history of neck and shoulder pain, muscle atrophy, and weakness of the left arm and hand was referred to our department. Physical examination revealed weakness and numbness of the left upper limb and balance problems causing ambulation difficulty, particularly when going downstairs. Her neurological function was classified as grade D on the AIS and grade II on the MMS. MRI of the cervical spine revealed an intramedullary tumor with high signal intensity on T2-weighted image (T2WI) (Figure 1A,C,D,E), extending from the medulla oblongata to the C5 level, accompanied by extensive edema. The tumor was located disproportionately on the left side of the spinal cord without contrast enhancement on gadolinium-enhanced T1WI (Figure 1B). A high uptake area, corresponding to the tumor on MRI, was observed with 11C-methionine positron emission tomography (PET) (Figure 2A); however, it was not described as a high-uptake lesion on 18F-fluoro-deoxy-glucose PET (Figure 2B). Surgical treatment for tumor removal was performed via the posterior approach under transcranial motor and sensory-evoked potential monitoring. The tumor was resected through the left dorsal root entry zone following suboccipital craniotomy and laminoplasty from C1 to C5. The cervical spinal cord tumor was removed almost completely; however, the tumor in the medulla oblongata was removed only partially (Figure 3A) because of the unclear interface between the tumor and normal brainstem parenchyma. The neurological symptoms were transiently worsened to grade III on MMS but not changed grade D on AIS at one week after surgery.

Sanger sequencing using DNA derived from tumor tissues revealed an *IDH1* R132C mutation. These results showed that the integrated diagnosis of this tumor was diffuse astrocytoma, IDH-mutant, based on the 2016 WHO classification of central nervous system tumors. Postoperatively, her ambulation improved, although her left arm numbness and weakness remained unchanged. The patient received focal radiotherapy (a dose of 48 Gy delivered in 24 fractions) for the residual medulla oblongata tumor. There was no tumor regrowth on T2WI MRI 14 months after surgery (Figure 3B). Her ambulation improved as almost same status of pre-operation but her left arm numbness and weakness remained unchanged in an outpatient follow-up.

The patient was a 14-year-old boy (Case #6) and was referred to our hospital because of headache, nausea, and rapid muscle weakness in his extremities within 3 months of the onset of the symptoms. Physical examination revealed tetraparesis. His lower limbs could reach the full range of motion against gravity, but it was impossible to maintain a standing position. The patient was classified as grade D on the AIS and grade IV on the MMS. An MRI of the cervical spine revealed an intramedullary tumor with low signal intensity on T1WI (Figure 4A) and high signal intensity on T2WI (Figure 4B), extending from the medulla oblongata to the C6 level. The tumor showed intense homogeneous enhancement on gadolinium-enhanced T1WI from the medulla oblongata to the C4 level (Figure 4C). Both 11C-methionine PET and 18F-fluoro-deoxy-glucose PET showed a high uptake area, corresponding to the tumor on the MRI (Figure 5). Surgical treatment for tumor removal was performed via the posterior approach under transcranial motor and sensory-evoked potential monitoring. The tumor was resected through the dorsal median sulcus of the spinal cord following suboccipital craniotomy and laminoplasty from C1 to C4. The tumor was fragile, and the boundary between the tumor and the normal spinal cord was unclear; thus, the tumor could be removed only partially. Sanger sequencing using DNA derived from tumor tissues revealed an *H3F3A* K27M mutation. These results showed that the integrated diagnosis of this tumor was anaplastic astrocytoma, *H3F3A* K27M-mutant, based on the 2016 WHO classification of CNS tumors. Postoperatively, his neurological status was worsened on AIS to grade C but remained unchanged on MMS grade IV. MRI revealed residual tumor (Figure 6A) and the patient received focal radiotherapy (a dose of 54.0 Gy delivered in 30 fractions) with concomitant and adjuvant temozolomide and bevacizumab. Two months after the first surgery, a syringosubarachnoid shunt, owing to the expanding cyst and for tumor re-excision, was performed via C5 and C6 laminoplasty (Figure 6B). Nevertheless, residual tumor tissue was still present, and adjuvant therapy was continued (Figure 6C). Despite these treatments, the patient’s neurological symptoms did not improve and gradually worsened (Figure 6D). Eventually, he died 18 months after the first surgery.

### 3.2. Types of Tumor Progenitor Cells Associated with Functional and Survival Outcomes in Spinal Gliomas

In addition to Table 1 and Table 2, Figure 7 illustrates the initial tumor locations, postoperative residual tumor parts, and the time-course changes in the MMS in each case. Cases #1–3 were pathologically diagnosed as WHO grade I pilocytic astrocytoma based on evidence of a biphasic pattern of glial fibrillary acidic protein immunostaining and the existence of Rosenthal fibers. Pilocytic astrocytomas are relatively well demarcated (Appendix A). As in Cases #1 and #2, tumor removal could be achieved up to 50–90% (Table 1). In these cases, MMS did not deteriorate at 1 year after surgery. In Case #2, neurological function transiently worsened from MMS grade III to grade IV due to surgical damage, but eventually recovered to grade II, 1 year after surgery. The AIS remained at grade D for the follow up period. The tumor in Case #3 was a chronic lesion in which the preoperative function had already been aggravated to grade IV on the MMS and grade C on the AIS. Thus, a minimally safe resection was performed so that the patient would not experience further corruption of neurological function.

Cases #4 and #5 were WHO grade II astrocytomas, *IDH1*-mutant. This tumor type has an infiltrating nature to the surrounding spinal cord, precluding complete resection. Indeed, in these cases, the extent of resection was only 50%. However, the neurological outcomes were relatively stable. In Case #4, neurological function transiently worsened from MMS grade II to grade III due to surgical damage, but the neurological symptoms improved to almost the same state as before surgery, 1 year after surgery. The AIS remained at grade D for the follow up period. In Case #5, numbness in both hands newly appeared due to the surgical injury, but there was no deterioration in motor function. The neurological symptoms remained stable at one year postoperatively.

In contrast, in the cases of astrocytoma (Cases #6–8), H3F3A K27M discriminated strikingly from the aforementioned types. This aggressive tumor type always requires adjuvant therapy consisting of radiotherapy and chemotherapy. Nevertheless, the tumor is life threatening. In our cohort, we lost all patients with the *H3F3A* K27M mutation within 5 years. The neurological function evaluated by the MMS was devastating over time in each case.

## 4. Discussion

Compared to studies on intracranial gliomas, those on rare spinal cord gliomas have been limited, and the molecular characteristics of spinal cord gliomas remain largely unknown [16]. Moreover, to the best of our knowledge, there are no reports describing the association of progenitor cells with driver mutations and tumor-associated spinal cord injury. In particular, owing to the rarity of occurrence of spinal cord tumors with *IDH* mutations, the association between this gene mutation and clinical features has never been described. In this study, we evaluated the relationship between genetic mutations and the degree of deterioration of neurological status in eight patients with spinal cord gliomas. We found three clinical entities divided in terms of the severity of tumor-associated spinal cord injury: WHO grade I pilocytic astrocytoma, grade II astrocytoma with *IDH* mutation, and grade III/IV astrocytoma with *H3F3A* K27M mutation.

### 4.1. Grade I Pilocytic Astrocytoma

Supratentorial pilocytic astrocytomas are more likely to harbor the *BRAF* V600E mutation, whereas posterior fossa and spinal cord pilocytic astrocytomas frequently harbor *BRAF-KIAA1549* fusion oncogenes [8]. A study of 17 spinal cord astocytomas revealed that 80% of grade I astrocytomas harbored mutations in the *BRAF* genes, with 40% harboring *BRAF-KIAA1549* translocation and the other 60% harboring a *BRAF* copy number gain. The findings of different genetic underpinnings between intracranial and spinal cord astrocytomas have an important implication that spinal pilocytic astrocytoma may be a distinct entity arising from unveiled tumorigenic mutations. Therefore, mutant progenitor cells with invasive and infiltrative nature may not be present in this tumor type.

### 4.2. Grade II Astrocytoma

Grade II spinal astrocytomas appear to harbor *BRAF-KIAA1549* translocations and *BRAF* amplifications [8]. There is limited information regarding spinal grade II astrocytomas. However, in this study, we found *IDH1* R132C and R132H mutations in two grade II tumors. An *IDH* mutation was first identified in an intracranial glioblastoma in 2008 [17]. It has become clear that *IDH* mutations in intracranial gliomas are associated with low malignancy [18,19,20,21,22]. In most cases, the mutation occurs in *IDH1* codon 132 or *IDH2* codon 172. Approximately 90% of *IDH* mutations are *IDH1* R132H [18,19,23]. Therefore, immunohistochemistry using an antibody against IDH1 R132H is the most widely used method for detecting IDH mutations.

To date, only three reports have described *IDH* mutations in spinal gliomas. None of these referred to *IDH1* R132C mutations [7,17]. In 2017, Takai et al. [7] first reported a spinal cord astrocytoma, *IDH1* R132S mutation. Five cases of spinal gliomas with *IDH* mutations were described by Konovalov et al. [17], in which two cases had *IDH1* R132H mutations, one had the *IDH1* R132G mutation, and the remaining two cases had unique substitutions at positions 82 (Arg→Lys, R82K) and 76 (Ile→Thr, I76T) of the *IDH1* gene. These analyses suggested that the frequency of *IDH* mutants found in spinal gliomas might be distinct from those found in intracranial gliomas, where the *IDH1* R132H mutation is the most common (90% or more). Although immunohistochemistry using an anti-IDH1 R132H antibody is useful as a screening test for the detection of IDH mutations in intracranial gliomas [24], Sanger sequencing and ddPCR of the *IDH1* or *IDH2* mutation are also required, particularly in spinal gliomas. Whether identification of *IDH* mutations can be a good prognostic factor for spinal gliomas is inconclusive to date. The patients in the present study were in good condition for 22 and 37 months following partial tumor resection. Grade II lesions can be more infiltrative than grade I lesions but with a slow growth pattern [25]. In many cases, these are poorly enhancing but diffuse in nature and have indistinct borders. In this study, the time course of neurological status was stable during the follow-up period. This type should be in the intermediate group based on neurological function. However, similar to intracranial astrocytomas, low-grade lesions may transform into high-grade lesions, but the underlying molecular mechanisms are unclear. Therefore, further follow-up is warranted.

### 4.3. Grade III/V Astrocytoma

The most important gene in spinal cord astrocytomas is histone 3 variant H3.3 (*H3F3A*), which has been implicated in the tumorigenesis of both intracranial and spinal astrocytomas [26,27]. The *H3F3A* K27M mutation has predominantly been detected in malignant astrocytomas arising in structures of the midline, including the thalamus, brainstem, and spinal cord, and as such is listed as a separate entity in the 2016 WHO classification [28,29]. Comparison of low-grade (grades I and II) and high-grade (grades III and IV) spinal cord astrocytomas showed a preponderance of the K27M mutation in grades III and IV spinal cord astrocytomas. Grade III or IV spinal cord astrocytomas lack distinct borders and are very likely to recur rapidly [30,31,32]. The *H3F3A* mutation is segregated from the *IDH1* mutation. Progenitor cells with *H3F3A* mutation aggressively destroy the spinal cord, causing severe neurological deficits and poor prognosis. In our series, all patients with grade III/IV astrocytoma and H3F3A K27M died along with worsened neurological function within 60 months.

## 5. Conclusions

The spinal cord is a vital structure that connects the whole body to the brain. Neurological assessments, in addition to evaluation of the prognosis of spinal cord tumors, should be considered depending on the type and location of the tumor. In this study, grade III and grade IV gliomas clearly have a poor functional prognosis, 1 year after surgery, and grade II gliomas with IDH mutation have better functional prognosis similar to grade I, 1 year after surgery. However, it remains unknown what mutations affect functional prognosis and life expectancy in grade I/II glioma groups in the present cohort. Although spinal gliomas are extremely rare, further long-term studies are warranted to determine the differences in functional prognosis among their genetic types.

## Figures and Tables

**Figure 1 cells-10-02525-f001:**
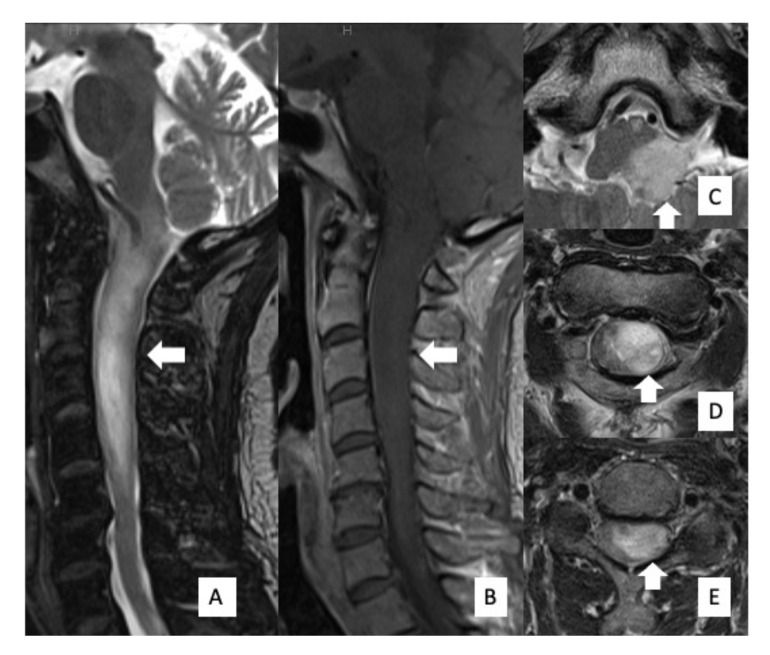
Preoperative MRI findings. MRI showed an intramedullary mass lesion, extending from the medulla oblongata to C5 (arrow). High signal intensity was evident on T2WI (**A**) and low signal intensity was evident on T1WI (**B**). Axial images of T2WI showed that the tumor was disproportionately located on the left side of the spinal cord (**C**–**E**).

**Figure 2 cells-10-02525-f002:**
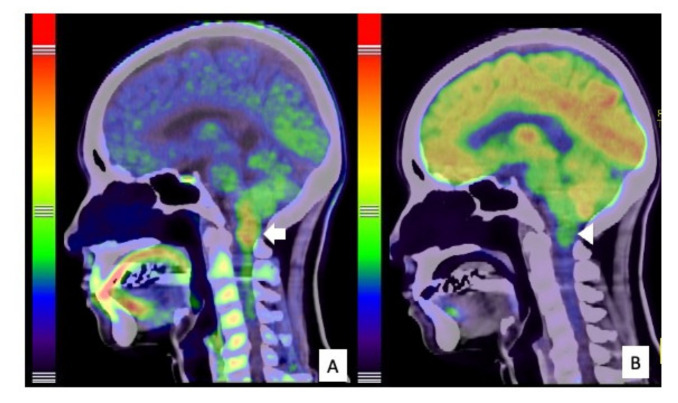
PET findings of the tumor. ^11^C-Methionine PET showed high uptake by the tumor (arrow), particularly from the medulla oblongata to the spinal cord at the C1 level, indicating spinal astrocytoma (**A**). 18F-Fluoro-deoxy-glucose (FDG) PET showed generally low FDG uptake by the tumor (arrowhead), indicating low malignancy (**B**).

**Figure 3 cells-10-02525-f003:**
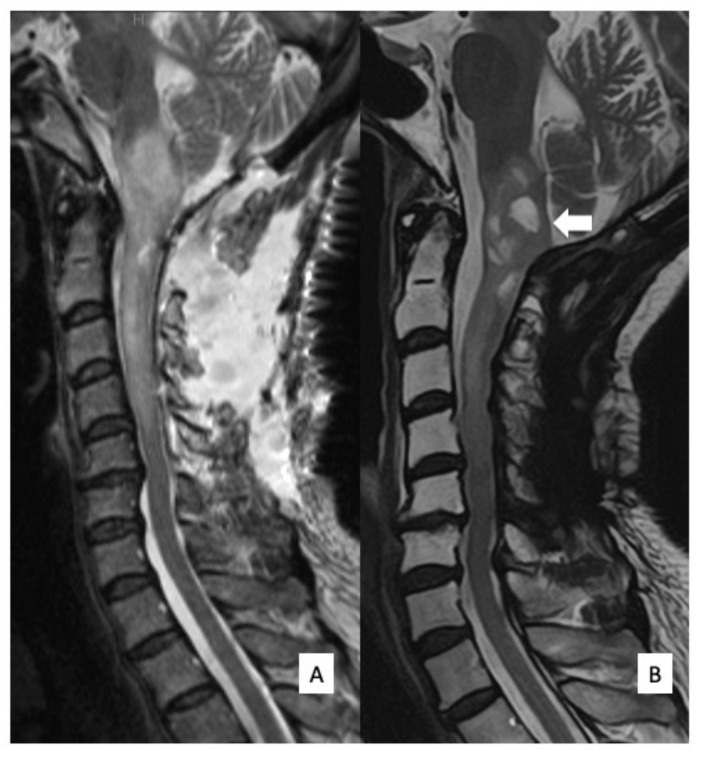
Postoperative MRI findings. T2WI sagittal MRI obtained a week after surgery (**A**) and 12 months after surgery (**B**). At 12 months after surgery, when the patient had already undergone postoperative radiation therapy, the tumor demonstrated cystic degeneration in the medulla oblongata (arrow), and no tumor regrowth was observed.

**Figure 4 cells-10-02525-f004:**
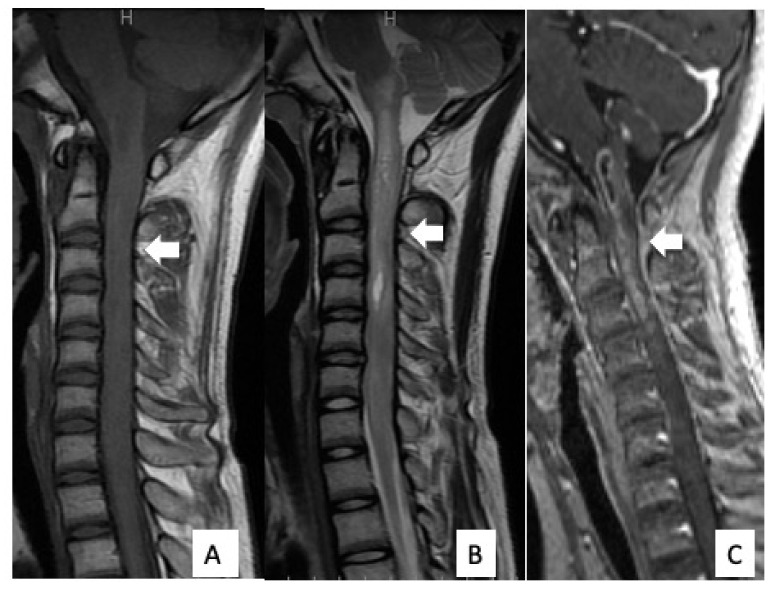
Preoperative MRI findings. MRI revealed an intramedullary tumor extending from the medulla oblongata to the cervical spine (arrow). There was low signal intensity on T1WI (**A**), high signal intensity on T2WI (**B**), and contrast enhancement on gadolinium-enhanced T1WI (**C**).

**Figure 5 cells-10-02525-f005:**
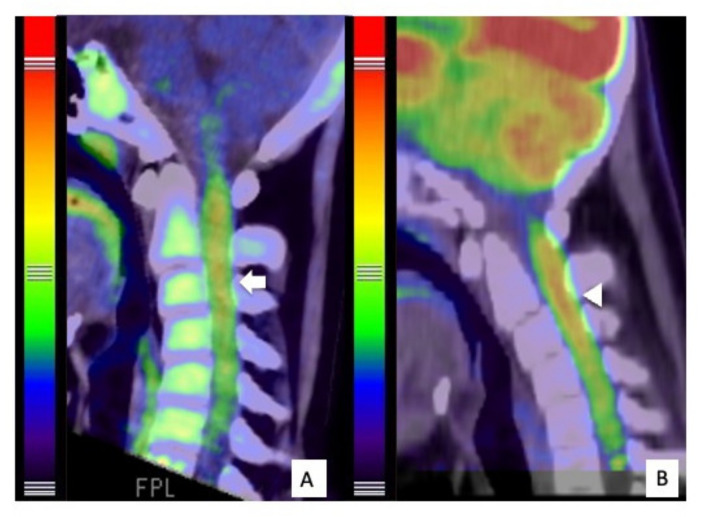
PET findings of the tumor^. 11^C-Methionine PET revealed high uptake by the tumor (arrow), indicating spinal astrocytoma (**A**). 18F-Fluoro-deoxy-glucose positron emission tomography (18F-FDG PET) showed high uptake by the tumor (arrowhead), particularly in the spinal cord at the C1-C3 levels (**B**).

**Figure 6 cells-10-02525-f006:**
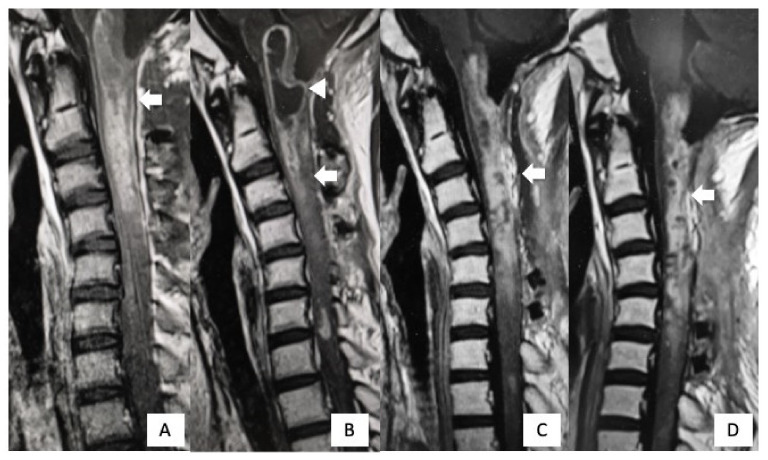
Postoperative MRI findings. Gadolinium-enhanced T1WI imaging 1 week after surgery revealed a residual tumor (arrow). (**A**) Two months after the first surgery, cystic lesions appeared from the medulla oblongata to C2 (arrowhead). (**B**) After syringosubarachnoid shunt for the expanding cyst and tumor re-excision, the cystic lesion disappeared, but the residual tumor was still present (arrow). (**C**) One year after the first surgery, there was no further regrowth of the residual tumor (arrow) (**D**).

**Figure 7 cells-10-02525-f007:**
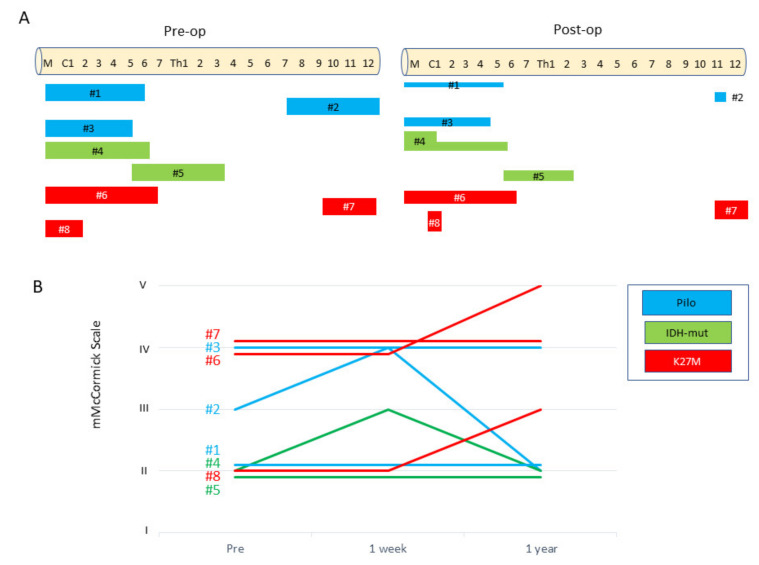
The clinical course of patients with spinal cord glioma comprised three groups: WHO grade I (pilocytic astrocytoma), shown in blue; WHO grade II (with *IDH* mutant), shown in green; and WHO grade III/IV (with *H3F3A* mutant), Scheme 1. week and 1 year after surgery. One year after surgery, the functional status of WHO grade I and II gliomas remained unchanged or improved, compared with that before surgery, while that of grade III/IV gliomas remained worsened or unchanged, and the severity of the scale was higher.

**Table 1 cells-10-02525-t001:** Summary of clinical features, WHO grades, and genetic statuses.

Case	Age	Sex	Location	Diagnosis	WHO Grade	Genetic Status	EOR	OS (months)	Dead/Alive
1	29	M	M-C5	Pilocytic astrocytoma	1	-	50%	66	A
2	19	M	Th8-12	Pilocytic astrocytoma	1	-	90%	34	A
3	48	F	M-C4	Pilocytic astrocytoma	1	-	30%	62	A
4	53	F	M-C5	Astrocytoma	2	IDH-mut	40%	22	A
5	42	F	C5-Th3	Astrocytoma	2	IDH-mut	60%	37	A
6	14	M	M-C6	Glioblastoma	4	H3F3A-mut	5%	18	D
7	62	F	Th10-12	Anaplastic astrocytoma	3	H3F3A-mut	30%	22	D
8	15	M	M-C1	Anaplastic astrocytoma	3	H3F3A-mut	60%	57	D

M, male; F, female; M, medulla oblongata; C, cervical spine; Th, thoracic spine; mut, mutant; EOR, extent of resection; OS, overall survival time from diagnosis.

**Table 2 cells-10-02525-t002:** Neurological status of patients.

Case	WHO Grade	Neurological Findings								
		Pre-op			1 wk-Post-op			1 yr-Post-op		
		Neurological Status	AIS	MMS	Changes in Neurological Status	AIS	MMS	Changes in Neurological Status	AIS	MMS
1	WHO grade I	weakness in left upper limb (4/5), numbness in left upper & lower limb	D	II	unchanged	D	II	unchanged	D	II
2		weakness in the limbs (4/5), bilateral femoral pain, urinary retention	D	III	worsend (Left lower limb muscle strength deteriorated to 2-3/5, hypesthesia in both lower limbs, urinary retention)	D	IV	improved (Almost normal lower limb muscle strength, hypesthesia in left lower limbs, urination without catheterization)	D	II
3		weakness in the limbs (right 2/5, left 3/5)	C	IV	unchanged	C	IV	slightly improved (weakness in both limbs (right 2/5, left 3-4/5))	C	IV
4	WHO grade II	weakness in left proximal upper extremity muscle (4/5), left hand numbness, sensory ataxia	D	II	worsened (weakness in left upper limb (3/5) and lower limb (4/5), numbness in the limbs, sensory ataxia)	D	III	improved to the almost same status as pre-op	D	II
5		weakness in right lower limb (4/5), numbness in both lower limbs	D	II	worsened (weakness in right lower limb (4/5), numbness in the limbs)	D	II	unchanged from 1 wk-post-op status	D	II
6	WHO grade III/IV	weakness in the limbs (2-3/5)	D	IV	worsened (weakness in upper limbs (3/5) and lower limbs (2/5))	C	IV	worsend from 1 wk-post-op staus (tetraparesis, sensory disorders of the extremities, ureteral catheterization)	C	V
7		weakness in both lower limbs (right 2/5, left 0/5), Sensory disorders below Th12 level, urinary retention	C	IV	unchanged	C	IV	unchanged	C	IV
8		weakness in the limbs (4/5)	D	II	unchanged	D	II	worsend from 1 wk-post-op staus (weakness in the limbs (2-3/5))	D	III

AIS, American Spinal Cord Injury Association impairment scale; MMS, Modified McCormick Scale. Muscle strength was described on a scale based on manual muscle testing (MMT).

## Data Availability

Datasets were generated during the study. We endorsed MDPI Re-search Data Policies.

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
