# Peer review of "Driver Genetic Mutations in Spinal Cord Gliomas Direct the Degree of Functional Impairment in Tumor-Associated Spinal Cord Injury"

_cells, 2021, doi:10.3390/cells10102525_

Round 1

Reviewer 1 Report

Specific comments:

  • The abstract did not match the title of “spinal cord glioma progenitor cells” – nothing was mentioned in the abstract. The authors need to stick with one set of terminology. How did they determine “spinal cord glioma progenitor cells?” What were assays? Any non-progenitor controls? If they wanted to publish in Journal Cells, they need to provide cellular characterization data sets.
  • Fig 1 – Fig 6 should be provided with arrowheads and scale bars. All of the Figures should be aligned with Table 1, illustrating which image is for which patient.
  • Lines 204-207 “Cases #1–3 were pathologically diagnosed as WHO grade I pilocytic astrocytoma based on evidence of a biphasic pattern of glial fibrillary acidic protein immunostaining and the existence of Rosenthal fibers” – micrographs should be provided here to align with Table 1.
  • Fig 7 should be arranged to have a side-by-side comparison of pre- with post-op with scale nearby (corresponding to each patient). The panel B was with poor resolution, hard to read, even worse than that the number did not align with the case curve. The panel B patterns of Case #2 and Case #4 did not match the scale change patterns (Table 1).
  • Case #2, Table 1, why did the modified McCormick Scale (MMS) shift from III to IV? So did case #4 from II to III? The panel B patterns of Case #2 and Case #4 did not match the scale change patterns (Table 1).
  • Fig 2 & 5 should be provided with a color scale explanation followed by arrowheads of interest.
  • Lines 236 – 237: 5 citations clustered in one location, which did not tell the reader what was about each citation - “the molecular characteristics of spinal cord gliomas remain largely known [15-20].”
  • Lines 241-242 “the functional and basic molecular pathological…” – the authors added a layer of confusion for their topic of focus without any data on either functional or molecular pathology.
  • Lines 259 – 260 “found IDH1 R132C and R132H mutations in two grade…” Where did the authors’ data sets?
  • Lines 89-90: “Immunohistochemistry for the anti-IDH1 R132H antibody” – where was the data set?
  • Line 90: “a marginal positivity in Case #5” – what was the data? The controls?
  • Lines 301 - 307, “Conclusion” did not directly reflect on their own data sets, but another section regurgitated sentences of the introduction. They need to extract their thoughts and envision the future.

Author Response

To Reviewer 1:

[Comment 1] The abstract did not match the title of “spinal cord glioma progenitor cells” – nothing was mentioned in the abstract. The authors need to stick with one set of terminology. How did they determine “spinal cord glioma progenitor cells?” What were assays? Any non-progenitor controls? If they wanted to publish in Journal Cells, they need to provide cellular characterization data sets.

Reply: Thank you for your feedback. We have changed the title to “Driver genetic mutations in spinal cord gliomas direct the degree of functional impairment in tumor-associated spinal cord injury.” “Clonal Evolution of Cancers” suggests that the mutation of IDH1/2 or H3F3A could be the driver or truncal genetic alterations that induce tumorigenesis and trigger subsequent (epi)genetic events. Mutations in IDH1/2 or H3F3A were found in all tumor cells. Therefore, the very first single cell that undergoes mutation is regarded as a tumor progenitor cell.

[Comment 2] Fig 1 – Fig 6 should be provided with arrowheads and scale bars. All of the Figures should be aligned with Table 1, illustrating which image is for which patient.

Reply: The arrowheads have been provided to indicate the region of interest. The case numbers have been provided in the Figure legends.

[Comment 3] Lines 204-207 “Cases #1–3 were pathologically diagnosed as WHO grade I pilocytic astrocytoma based on evidence of a biphasic pattern of glial fibrillary acidic protein immunostaining and the existence of Rosenthal fibers” – micrographs should be provided here to align with Table 1.

Reply: As per your suggestion, the micrographs of Case #2 are provided in Supplementary Figure 1, which illustrates a biphasic pattern of glial fibrillary acidic protein immunostaining and the existence of Rosenthal fibers.

[Comment 4] Fig 7 should be arranged to have a side-by-side comparison of pre- with post-op with scale nearby (corresponding to each patient). The panel B was with poor resolution, hard to read, even worse than that the number did not align with the case curve. The panel B patterns of Case #2 and Case #4 did not match the scale change patterns (Table 1).

Reply: As suggested, Panel A has been shown in a side-by-side manner. Instead, panel B was made larger so that it could be more visible. In Table 2, the modified McCormick Scale (MMS) at 1 year after surgery has been added, so panel B matches the scale change pattern.

[Comment 5] Case #2, Table 1, why did the modified McCormick Scale (MMS) shift from III to IV? So did case #4 from II to III? The panel B patterns of Case #2 and Case #4 did not match the scale change patterns (Table 1).

Reply: This is due to a temporary worsening of neurological function. We would like to describe that functional recovery is more likely to occur in non-K27M tumors. We have revised the Results section as follows: “In Case #2, neurological function transiently worsened from MMS grade III to grade IV due to surgical damage, but eventually recovered to grade II, 1 year after surgery.” In case #4, muscle weakness in the right upper limb and sensory disturbance in the extremities appeared transiently due to surgical damage, but the neurological symptoms improved to almost the same state as before surgery, 1 year after surgery.”.

[Comment 6] Fig 2 & 5 should be provided with a color scale explanation followed by arrowheads of interest.

Reply: We have revised the figures as per your suggestions.

[Comment 7] Lines 236 – 237: 5 citations clustered in one location, which did not tell the reader what was about each citation - “the molecular characteristics of spinal cord gliomas remain largely unknown [15-20].”

Reply: Instead of five citations, we have cited one review article “Abd-El-Barr, M. M.; Huang, K. T.; Moses, Z. B.; Iorgulescu, J. B.; Chi, J. H., Recent advances in intradural spinal tumors. Neuro Oncol 2018, 20, (6), 729-742.”

[Comment 8] Lines 241-242 “the functional and basic molecular pathological…” – the authors added a layer of confusion for their topic of focus without any data on either functional or molecular pathology.

Reply: We have revised the corresponding passage as follows: “In this study, we evaluated the relationship between genetic mutations and the degree of deterioration of neurological status in eight patients with spinal cord gliomas.”

[Comment 9] Lines 259 – 260 “found IDH1 R132C and R132H mutations in two grade…” Where did the authors’ data sets?

Reply: We have provided the data in the “Information for Reviewers Only” to show IDH1 R132C and IDH1 R132H (see the attachment file).

[Comment 10] Lines 89-90: “Immunohistochemistry for the anti-IDH1 R132H antibody” – where was the data set? Line 90: “a marginal positivity in Case #5” – what was the data? The controls?

Reply: In clinical pathology, we do not stain controls every time. Because immunohistochemistry for the anti-IDH1 R132H antibody in Case #5 was ambiguous, we further confirmed the positivity of IDH1 R312H by droplet digital PCR. The data are shown in the “Information for Reviewers Only” (see the attachment file).

[Comment 11] Lines 301 - 307, “Conclusion” did not directly reflect on their own data sets, but another section regurgitated sentences of the introduction. They need to extract their thoughts and envision the future.

Reply: We have revised the “Conclusion” as follows: The spinal cord is a vital structure that connects the whole body to the brain. Neurological assessments, in addition to evaluation of the prognosis of spinal cord tumors, should be considered depending on the type and location of the tumor. In this study, grade III and grade IV gliomas clearly have a poor functional prognosis, 1 year after surgery, and grade II gliomas with IDH mutation have better functional prognosis similar to grade I, 1 year after surgery. However, it remains unknown what mutations affect functional prognosis and life expectancy in grade I glioma groups in the present cohort. Although spinal gliomas are extremely rare, further long-term studies are warranted to determine the differences in functional prognosis among their genetic types.

Reviewer 2 Report

The article is very interesting and could help to establish evolutionary prognoses in spinal gliomas.

This work focuses on the study of the possible associations of certain gene mutations with a worse vital prognosis or the malignancy of gliomas when involving the spinal cord. In this sense, the different radiological findings are described according to the pathological analyzes of the glioma extracted and a greater or lesser probability of malignant evolution, inability to completely remove the tumor or worse clinical evolution is attributed according to genetic findings. Since it is an extremely rare disease, it can be admitted to make these associations even if the sample of each genetic alteration is very few cases.

The article is well structured in terms of showing that the study of the existence of certain mutations in the glioma can help in the adequate follow-up and treatment (surgical and adjuvant).

However, the authors explain that the neurological assessment was carried out with the Modified McCormick Scale. This is a very poor scale to assess a functional evolution because it only takes into account the ability to walk. Spinal cord injury is a set of symptoms that is routinely assessed by exploring strength and sensitivity using motor balance scales and key sensory points (International Standards for Neurological Classification - ASIA Impairment Scale), the verification of medical complications (bladder , bowel, breathing) and the abilities that the patient maintains to carry out activities of daily living and other functions. The scale most used to monitor whether there are functional changes due to the natural evolution or due to the treatments received is the spinal cord injury independence scale (SCIM III). In the patients described, the authors say that they have upper limb involvement, so whether the resulting disability changes or not cannot be measured with a gait scale. To assess changes in walking ability, there is even a more sensitive scale, which is the walking index.in spinal cord injury (WISCI II)

Therefore, in order for the article to be published, it is possible to eliminate the reference to the functional evolution measured with the Modified McCormick Scale or to review the medical records of the patients (if they have been evaluated in Neurology or especially in Physical Medicine and Rehabilitation Departments, they will have the neurological examination and SCIM III)

Author Response

To Reviewer 3:

Reply: In the revised Table 1, we have added weakened muscles by manual muscle testing (MMT), ASIA Impairment Scale, sensory impairments, and bladder complications, as well as the Modified McCormick Scale (MMS). As per your suggestion, we agree that MMS assesses only the ability to walk, but the scale has been wildly used in functional assessment of intradural spinal tumors, and it can classify AISA “C” and “D” more precisely.

Reviewer 3 Report

Interesting work describing the importance of genetic mutations in spinal cord gliomas. These tumors are among the most devastating to the spinal cord, so finding gene mutations may be a chance to find a treatment.

In my opinion, the article lacks a detailed description of the neurological condition of patients. The authors referred only to the functional scale of MMS. In my opinion, this requires improvement and a broader description.

In result section and conclusion there are no implications for clinical practice. What data on gene mutations may change in the treatment of patients. The authors must complete the conclusions. 

Author Response

To Reviewer 4:

[Comment 1] Interesting work describing the importance of genetic mutations in spinal cord gliomas. These tumors are among the most devastating to the spinal cord, so finding gene mutations may be a chance to find a treatment.

In my opinion, the article lacks a detailed description of the neurological condition of patients. The authors referred only to the functional scale of MMS. In my opinion, this requires improvement and a broader description.

Reply: In the revised Tables 1 and 2, we have added weakened muscles by manual muscle testing (MMT), ASIA Impairment Scale, sensory impairments, and bladder complications, as well as the Modified McCormick Scale (MMS). As per your suggestion, we agree that MMS assesses only the ability to walk, but the scale has been wildly used in functional assessment of intradural spinal tumors, and it can classify AISA “C” and “D” more precisely.

[Comment 2] In result section and conclusion there are no implications for clinical practice. What data on gene mutations may change in the treatment of patients. The authors must complete the conclusions. 

Reply: We have revised the “Conclusion” as follows: The spinal cord is a vital structure that connects the whole body to the brain. Neurological assessments, in addition to evaluation of the prognosis of spinal cord tumors, should be considered depending on the type and location of the tumor. In this study, grade III and grade IV gliomas clearly have a poor functional prognosis, 1 year after surgery, and grade II gliomas with IDH mutation have better functional prognosis similar to grade I, 1 year after surgery. However, it remains unknown what IDH mutation affects functional prognosis and life expectancy in grade I glioma groups in the present cohort. Although spinal gliomas are extremely rare, further long-term studies are warranted to determine the differences in functional prognosis among their genetic types.

Round 2

Reviewer 1 Report

accepted.

Reviewer 2 Report

The text has been improved with the addition of AIS impairment scale.